# COVID-19-Associated Erythema Nodosum Detected on FDG PET/CT

**DOI:** 10.3390/diagnostics13030444

**Published:** 2023-01-26

**Authors:** Nora Eberfalvi Lipositsne, Kirsten Bouchelouche

**Affiliations:** Department of Nuclearmedicine and PET Centre Aarhus University Hospital, 8200 Aarhus, Denmark

**Keywords:** 18F-FDG-PET/CT, COVID-19, erythema nodosum

## Abstract

We report the case of a 69-year-old woman who underwent 18F-FDG PET/CT due to prolonged fever. One month before, the patient was diagnosed with COVID-19 infection. The 18F-FDG PET/CT showed several subcutaneous nodules with 18F-FDG uptake on the thorax and upper extremities and bilateral lung infiltrates due to organizing pneumonitis. Clinical examination revealed multiple tender nodules on thorax, arms, and legs, consistent with erythema nodosum (EN) induced by COVID-19 infection. The woman was treated with prednisone with a good effect on EN. To our knowledge, this is the first report on EN secondary to COVID-19 infection diagnosed on 18F-FDG PET/CT.

**Figure 1 diagnostics-13-00444-f001:**
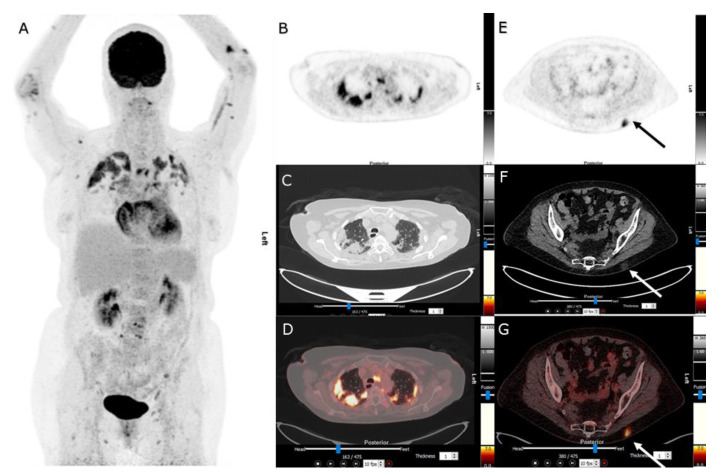
A 69-year-old woman underwent 18F-fluorodeoxyglucose positron emission tomography/computer tomography (18F-FDG PET/CT) as a part of the diagnostic work-up due to prolonged fever not responding to antibiotics. One month before, the patient was diagnosed with SARS-CoV-2 infection (COVID-19), confirmed via reverse transcription-polymerase chain reaction (RT-PCR). The 18F-FDG PET/CT scan showed high 18F-FDG uptake in multiple bilateral lung infiltrates, as can be seen on maximum intensity projection (MIP) image (**A**), axial PET (**B**), corresponding axial CT (**C**), and axial PET/CT slices (**D**). The lung infiltrates were caused by COVID-19 infection, as presented on 18F-FDG PET/CT [1]. Furthermore, 18F-FDG PET/CT showed several subcutaneous nodules on thorax and on upper extremities, as shown on MIP (**A**), axial PET (**E**), corresponding axial CT (**F**), and axial PET/CT slices (**G**). Clinical examination showed multiple tender erythematous nodules distributed on thorax, arms, and legs. These findings were consistent with the clinical manifestation of erythema nodosum (EN) [2]. Due to a high clinical index of suspicion of EN, no skin biopsy was performed. The woman initially received 37.5 mg prednisone once daily. Response evaluation was performed via clinical examination and a CT scan four weeks after initiation of steroid therapy. Both the pulmonary and the cutaneous manifestations showed significant remission, and the woman’s symptoms improved. Glucocorticoids are standard therapy for organizing pneumonia seen as late phase of COVID-19 [3,4], and for treatment of EN if non-responding on non-steroidal anti-inflammatory drugs (NSAIDs). The diagnosis of organizing pneumonitis and EN caused most likely by COVID-19 infection was established.

COVID-19 is best known for causing symptoms of upper respiratory tract infection in mild cases and fulminant pneumonia in severe disease, typically manifesting in ground glass opacity on chest imaging [5]. However, other clinical presentations such as gastrointestinal (anorexia, diarrhea, and vomiting), neurologic (fatigue, concentration issues, and “brain fog”), cardiac (myocarditis and dysrhythmias), and hematologic disorders (cytokine storm and thrombosis) have been reported as well [4,6,7], mostly diagnosed by symptoms, rather than imaging. Furthermore, cutaneous lesions related to COVID-19 are also common given that the receptor of SARS-CoV-2 (ACE2) is expressed on keratinocytes usually manifesting as chilblain- or urticaria-like and vesicular lesions [8], though EN is considered as a unique finding.

Erythema nodosum is a delayed-type hypersensitivity reaction mediated by the immune system and clinically presented as an acute septal panniculitis of the subcutaneous adipose lobule [2]. It can be either idiopathic or secondary to various underlying conditions, such infection, connective tissue disease, sarcoidosis, and lymphoma [2,9,10]. Normally, EN remains a self-limiting condition and usually resolves within a few weeks, although the exact etiology frequently remains unidentified and untreated. Erythema nodosum manifesting in cutaneous nodules may become noticeable when 18F-FDG PET/CT is performed in patients [11,12], and tuberculosis-induced EN on 18F-FDG PET/CT has been reported [13]. Cutaneous manifestations of COVID-19 are increasingly described in the literature, including a few cases of EN [14,15,16]. However, to our knowledge, this is the first report on EN secondary to COVID-19 infection diagnosed on 18F-FDG PET/CT. Thus, EN associated with fever, cough, and shortness of breath may be secondary to COVID-19 infection, and similar EN may be detected on 18F-FDG PET/CT performed in COVID-19 patients.

## Data Availability

Not applicable.

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
