# Peer review of "COVID-19-Associated Erythema Nodosum Detected on FDG PET/CT"

_diagnostics, 2023, doi:10.3390/diagnostics13030444_

Round 1
Reviewer 1 Report
The paper entitled “COVID-19 associated…. FDG PET/CT” is a short and well written case report. It is original because it is the first description of the FDG PET/CT aspect of a case of erythema nodosum related to COVID 19. The length of the paper is correct, it is easy to follow. The Discussion section is logically constructed. The English is correct, there is no spelling error. The references are OK. The images are of good quality.
In conclusion, the scope of this short case report is limited but the paper is original and well written.
Reviewer 2 Report
I have a few minor comments regarding this interesting case report:
-It would be worthwhile adding, if available, a patient's picture showing the erythematous nodules related to EN.
-A follow-up lung CT image at the same level of the PET/CT image shown in Figure 1 would be needed to show remission of lung alterations.
Reviewer 3 Report
The author reported a case of a 69-year-old woman with erythema nodosum diagnosed by 18F-FDG PET/CT probably
secondary to COVID-19 infection.
The case presented is interesting.
However, I suggest improving it by developing the main topic better. A broader discussion of cases of cutaneous and non-cutaneous pathologies developed after COVID-19 infection and studied by nuclear medicine examination could be useful. Moreover, I also suggest being less definitive about whether erythema nodosum is due to COVID-19.
Reviewer 4 Report
"The lung infiltrates were caused by COVID-19 infection with typical presentation on 18F-FDG PET/CT " The most common lung involvement by COVID-19 is instead GGO [doi: 10.1007/s11604-020-01010-7.], this aspect should be better explained
Proposed an explanatory hypothesis for EN related to a previous COVID-19 infection: an immune-mediated reaction, for example?
